# Optimization of Thermo-Mechanical Fatigue Life for Eutectic Al–Si Alloy by the Ultrasonic Melt Treatment

**DOI:** 10.3390/ma15207113

**Published:** 2022-10-13

**Authors:** Meng Wang, Jianchao Pang, Xinfeng Liu, Jianqiu Wang, Yongquan Liu, Shouxin Li, Zhefeng Zhang

**Affiliations:** 1Institute of Metal Research, Chinese Academy of Sciences, Shenyang 110016, China; 2Institute of Corrosion Science and Technology, Guangzhou 510530, China; 3School of Materials Science and Engineering, Northeastern University, Shenyang 110819, China

**Keywords:** eutectic Al–Si alloy, thermo-mechanical fatigue, ultrasonic melt treatment, damage mechanism, fatigue life

## Abstract

The eutectic cast Al–Si alloys with excellent high-temperature and casting performance are widely used in engine pistons. During frequent starts and stops, the thermo-mechanical fatigue (TMF) is the most important failure cause. Ultrasonic melt treatment (UT) was chosen to compare and investigate the influence of micro-structures on fatigue life and damage mechanisms of as-cast (AC) eutectic Al–Si alloys under TMF loading. After UT, the grain size, primary Si, and intermetallic particles are reduced significantly in the alloy; fatigue life increases obviously. As a result of pilling-up of dislocations, the competitive effects of the critical strain/stress for fatigue crack nucleation can be found. There are two different crack initiation mechanisms under TMF: one is primary Si fracture for AC alloys with limited critical strain/stress for fatigue crack nucleation at fractured Si particles, and the other is primary Si debonding for UT alloys with increasing critical fracture strain/stress. After the crack initiation, the fractured or debonded primary phases provide the advantages for the further development of main cracks for both alloys. The UT alloy (805 ± 253 cycles) has about twice the TMF life of the AC alloy (403 ± 98 cycles). The refinement of micro-structures is instrumental in improving the fatigue resistance and life of TMF for the UT alloy.

## 1. Introduction

Due to the high strength-to-weight ratio, outstanding thermal conductivity, castability, and high-temperature mechanical strength, the eutectic Al–Si cast alloys with multi-phases are used extensively in engine pistons [1]. According to the current development situation, the service temperature and mechanical load increase with the increasing diesel engine efficiency and peak cylinder pressure [2]. At the same time, the materials in some parts have reached their limit, and it has become difficult to further improve the temperature (more than 400 °C) and load (more than 20 MPa) [3]. The failure mechanism is the interaction of thermal/mechanical loading, resulting in thermo-mechanical fatigue (TMF) and subsequent failure, which is the most relevant factor affecting piston fatigue life. Therefore, it is significantly important to improve the TMF property of the alloys for the reliability of pistons [4].

In order to further improve the properties of the eutectic Al–Si alloys, alloy elements are constantly added, such as mainly adding Ni, Cu, Mg, and Mn, and the minor addition of Na, Sr, or Sc [5,6]. A high content of elements leads to complex phases in the alloy, including micron-scale primary phases (α-Al matrix, primary Si, eutectic Si, Mg_2_Si, AlCuNi, and so on) and nanoprecipitates (G.P. zone, θ(Al_2_Cu), θ′, and θ″). The transition elements Ni and Cu are regarded as the most useful elements in terms of the effect on the high temperature tensile behavior of the Al–Si piston alloy [7,8,9]. Although the high-temperature strength is improved monotonically, the fatigue performance increases first, and then decreases with the increase of Ni content [10]. Bugelnig et al. also found that the higher tensile strength at elevated temperatures can be derived from high volume fractions of highly interconnected rigid phases. However, the limited ductility reduces the TMF resistance [11]. At present, the content of Si, Ni, Cu, and other elements in this alloy has exceeded 20%, and the negative effects, such as the fatigue life reduction caused by the primary phase, begin to occur. It is difficult to further improve the high-temperature strength and fatigue life by increasing the content of alloying elements.

For Al–Si alloys, Huter et al. [4] found that there is little correlation between oxidation and durability of TMF, and the main fatigue crack is induced by hard phases. Floweday et al. [12] further pointed out that TMF crack initiation is caused by the fractured primary Si phase in the piston of an eutectic Al–Si alloy. Ultrasonic melt treatment is one of the most promising methods for refining primary phases and improving the mechanical properties at different temperatures [13,14]. The results show that the strength and plasticity of eutectic Al–Si alloys at high temperature are greatly improved after ultrasonic melt treatment with the refinement of primary Si and aluminides. The refinement of micro-structures can be derived from the cavitation-induced heterogeneous nucleation and dendrite fragmentation [15,16]. However, there are few studies on the effect of ultrasonic melt treatment on the elevated temperature fatigue behavior of Al–Si alloys [13], especially under TMF loading. To study and compare the TMF behaviors of Al–Si alloys after different cast processes, the cyclic behaviors, fracture mechanisms, and damage processes are discussed and analyzed in the present paper.

## 2. Experimental Procedures

### 2.1. Preparation Process and Micro-Structure Analysis

The ingots of eutectic Al–Si alloys were melted at a temperature of about 800 °C, and then the melt was placed in the titanium ultrasonic electrode at a temperature of 720–730 °C for 60 s. The two groups of as cast alloys without (AC) and with ultrasonic melt treatment (UT) were derived from the cast piston ingots after T6 heat treatment, and the reference chemical composition is shown in Table 1. The grain size distributions were measured by the electron backscatter diffraction (EBSD) technique. One of the typical damage characteristics and fracture morphology around the main crack after TMF was observed by scanning electron microscopy (SEM) JSM 6510. The micro-structures were studied by the 3D high-resolution X-ray micro-tomography (XRM) technique using the laboratory-based Xradia Versa XRM-500 system. The resolution of XRM is about 1 µm. More detailed information on the XRM program can be found in other references [17,18].

### 2.2. Thermo-Mechanical Fatigue Process

During the experiment, MTS810 servo hydraulic testing machine was used to realize thermo-mechanical loading. The samples were heated by induction heating coil, and cooled by compressed air. The relationship between the reference temperature (*T*_0_ = 25 °C), the testing temperature (*T*), total strain (ε_t_), mechanical strain (ε_m_), thermal strain (ε_th_), and the thermal expansion coefficient (α) of the eutectic Al–Si alloy can be written as [19,20]:(1)$$\varepsilon_{t}=\varepsilon_{\mathrm{th}}+\varepsilon_{m}=\alpha \left(T-T_{0}\right)+\varepsilon_{m}.$$

Before TMF test, the values of α and ε_th_ were measured under zero-load condition and the value is about 20.8 × 10^−6^/°C (Figure 1a). Axial total strain was measured with 25 mm high temperature extensometer. Since the thermal expansion is nonlinear by the characteristics of the material itself, the constraint factor (η) is defined as the ratio of the mechanical strain amplitude (ε_m_) to the thermal strain amplitude (ε_th_):(2)$$\eta =\frac{\varepsilon_{m}}{\varepsilon_{\mathrm{th}}}.$$

The value of constraint factor (η = 1) remains constant during the TMF tests in this study [21]. The standard TMF testing provides a method to simulate complex working load conditions with a definite phasing angle between thermal and mechanical loads. This can more accurately predict the fatigue performance of the piston under working conditions. The TMF cycle time is 182 s with a dwell time of 60 s at maximum temperature, and the cooling and heating rate are 5 °C/s. The temperature was measured and controlled by K-type thermocouple, attached to the center of the gauge length. The in-phase (IP) loading conditions of TMF cycle at temperatures varying from 120 to 425 °C and the mechanical strain (ε_m_) strain varying from 0.21% to 0.88% are shown in Figure 1b. Six samples were tested for each alloy, and the value of η remains constant during each TMF test (η = 1). The TMF tests specimens were machined to a parallel gauge section with a length of 30 mm and diameter of 8 mm.

## 3. Results and Discussion

### 3.1. Micro-Structure

Compact precipitation of various phases is the common feature of the alloy due to the high content of alloy elements (Si, Cu, Mg, Ni, Fe, and other transition elements). The alloy contains α-Al, primary/eutectic Si, and some intermetallic particles (such as Al_7_Cu_4_Ni, Al_3_CuNi, β-Al_5_FeSi, Mg_2_Si, and Al_3_ (Ti, Ce) phases). The differences in micro-structures influenced by the manufacturing process are discussed from two aspects: primary Si and grain structure. The primary Si particles are seriously agglomerated in the AC alloy (Figure 2a), but uniformly dispersed in the UT alloy with the smaller particles (Figure 2b). The size distribution curves of the primary Si particles in Figure 2e show that UT does not only greatly reduce the average size (from 33.1 μm to 24.3 μm), but also the maximum size (from about 65 μm to 55 μm). The EBSD data of the α-Al matrix indicate that the average grain size (from about 516 μm to 439 μm) and the maximum size (from 1273 μm to 1164 μm) decrease clearly for the UT alloy (Figure 2c,d,f). The heterogeneous nucleation induced by cavitation under ultrasonic agitation is most likely reason for the grain refinement and primary Si refinement.

In the eutectic Al–Si alloys, the intermetallic particles and Si play an important role in strengthening the ductile α-Al matrix. Nevertheless, one of the primary disadvantages of the Al–Si alloys is the complicated three-dimensional network of the hard phases. In order to investigate the distribution, inter-connectivity, and morphology, the three-dimensional micro-structures of intermetallic particles were observed with XRM technique (Figure 3). As shown in the tomographic reconstruction of AC and UT alloys, the different eutectic Si particles and intermetallic particles are disconnected with each other in 2D microstructures (Figure 3a,d). Nevertheless, these phases form a complicated interconnected 3D microstructure (Figure 3b,c,e,f), including different intermetallic particles (Al_7_Cu_4_Ni, Al_2_Cu, Al_3_CuNi, and Al_3_Ni phases).

### 3.2. Tensile and Isothermal Low-Cycle Fatigue Properties

These primary phases usually have a complex 3-dimensional structure and local connectivity (Figure 4a). The shrinkage pores in the alloy are usually present along with the presence of brittle intermetallic particles with complex geometries (Figure 4b–d). The shrinkage pores and lamellar Al_3_CuNi provide the preferred path for the fatigue crack initiation and crack extension (Figure 4e) [13,14]. It is worth emphasizing that interrupting the three-dimensional connection of UT alloys can obviously reduce the size of primary Si and intermetallic particles. The ultimate tensile strength (UTS) and elongation to fracture (EF) at different temperatures are displayed in Figure 5a. Due to the complex 3D microstructure of the primary phases, the damage mechanism is mainly derived from the primary phases fracture, and the alloy represents limit elongation at room temperature. With increasing temperature, both yield strength (YS) and ultimate tensile strength (UTS) decrease, and the elongation-to-fracture increases. For all the temperatures, both elongation and tensile strength-to-fracture of the alloys are increased for the UT alloy. The UTS rapidly decreases by about 84% when temperature rises from room temperature (290 MPa) to 425 °C (45 MPa). The primary cause is that when the temperature exceeds 300 °C, the strength of the age-hardenable Al alloy decreases rapidly with the strengthening precipitates coarsening (G-P zones and θ′ phases) [14,22]. Due to the high content of the hard phases in the alloy, the uniform deformation is limited, and the yield strength is similar to the tensile strength. Once the micro-structure containing larger size coarsening particles starts to form cracks or yield, the tendency for particles cracking and fracture increases dramatically. Notice that the yield strength (increases from 288 MPa to 324 MPa), tensile strength (increases from 290 MPa to 329 MPa), and elongation-to-fracture (increases from 0.52% to 0.87%) of the UT alloy at room temperature are significantly improved, which is primarily because of the refinement of micro-structures, including grain size, intermetallic particles, and primary Si. As for the 425 °C, the improvement in elongation-to-fracture (from 6.8% to 11.2%) can also be found. However, there is no significant change for tensile strength (from 45 MPa to 43 MP) for the UT alloy.

The dependence of isothermal low-cycle fatigue life (N_f_) on hysteresis energy (Ws) is expressed in the following form [23,24]:(3)$$D=\frac{1}{N_{f}}={\left(\frac{W_{s}}{W_{0}}\right)}^{\beta }.$$

The above equation can be transformed into the following equation:

(4)$$N_{f}={\left(\frac{W_{0}}{W_{s}}\right)}^{\beta }$$
where W_0_ and β are material parameters. The *N*_f_–*W*_s_ relationships on log–log scales show the intersection of isothermal fatigue life curves at high temperatures; the Ws here is hysteresis energy obtained from half-life time (Figure 5b). It is worth noting that the increase im temperature leads to the linear increase in W_0_ and 1/β, which are expressed as W_0_ = eT + f and 1/β = mT + n, respectively. The following new relation is gained by substituting the parameters of W_0_ and β in Equation (4):

(5)$$W_{s}=\left(\mathrm{eT}+f\right)\cdot N_{f}{}^{-\left(\mathrm{mT}+n\right)}.$$
where m, n, e, and f are constants of the given material and experimental condition. In addition, the evolution of fatigue life at constant W_s_ temperature is shown in Figure 5c. The N_f_ of the two alloys first increases and then decreases with the increase in temperature. The critical temperature (*T*_c_) occurs at the maximum fatigue life. For instance, the predicted *T_c_* of UT and AC alloys are about 275 °C and 325 °C, respectively (Figure 5c). In addition, the Tc of the UT alloy decreases, which means that the refining phases and prolonging fatigue life increase the proportion of high temperature time-dependent damage. The UT alloy (805 ± 253 cycles) has about twice the TMF life of the AC alloy (403 ± 98 cycles), as shown in Figure 5d.

### 3.3. Thermo-Mechanical Fatigue Behaviors

The information on the thermal strain and mechanical cyclic behavior is significant for understanding the damage mechanisms and life evolution of TMF. The representative cyclic deformation behaviors, including cyclic stress responses and half-life hysteresis curves, of the studied alloys under TMF are shown in Figure 6. The half-life hysteresis curves under TMF are unsymmetric and triangular, and differ from the symmetrical hysteresis curves of isothermal fatigue (Figure 6a,b). For both alloys, the tensile stress can be found in the process of cyclic heating. The tensile stress reaches the maximum value and then decreases with further increase in temperature and strain. As the heating process goes on, when the strain decreases, the compressive stress increases. The maximum tensile stress is lower than the maximum compressive stress because of the higher elastic modulus, yield strength, and flow stress at lower temperature. Therefore, the compressive stress is the average stress during the TMF cyclic loading process and the average stress is about 20~40 MPa. The main characteristic of cyclic stress is that the cycle stress is stabilized at low temperature (Figure 6c,d).

### 3.4. Thermo-Mechanical Fatigue Damage Mechanism

For the sake of researching the fatigue behaviors of the two alloys, it is unavoidable to study the relevant microscopic damage mechanisms. As shown in Figure 7, the fracture surfaces are overlapped with plastic deformation characteristics of the Al matrix, with some secondary cracks. In addition, the micro-structure damage characteristics around the fracture surface are also analyzed in Figure 8. During fatigue of both alloys, the fatigue micro-cracks usually originate from the fractured Si particles. These micro-cracks amalgamate with each other in the following cycles and propagate in the direction perpendicular to the load. The close-packed primary Si and intermetallic particles in the AC alloy are beneficial to the crack formation and propagation (Figure 7a). The fatigue damage around the main crack is mainly caused by the fractured Si particles (Figure 7b). Although the primary Si size is significantly reduced with UT, generally, the boundary of primary Si becomes the new nucleation position of the fatigue crack. Many micro-pores are formed at the interface of debonding Si, which plays a more significant role in the damage for the UT alloy (Figure 7c,d). Moreover, the cracks are smaller and the crack tip is circular. In order to analyze the distribution of the micro-cracks after TMF in the alloys, the 3D XRT images of the main crack can be found in Figure 9. It is found that the main cracks initiate from the surface of the specimen surface and propagate to the inside. Many micro-cracks are distributed around the main crack. The damage of the AC alloy is dominated by main cracks with more concentrated crack distribution. Due to the fracture of primary phases, some large-scale secondary cracks can be found. During the fatigue process, these micro-cracks formed by the concentrated broken primary phases merge and propagate rapidly, leading to the main crack and final failure (Figure 9a,b). However, the micro-cracks in the UT alloy are relatively uniformly distributed, and the main cracks are formed slowly. The micro-cracks in the alloy are difficult to propagate and merge to secondary cracks with large size (Figure 9c,d), which eventually shows a longer fatigue life (Figure 5d).

In addition to the primary phases, the nano-precipitation phases are another significant consideration affecting the TMF damage behaviors. Generally, the precipitation sequence of Al alloys containing Cu is supersaturated solution solid → Guinter–Preston (G–P) zones → θ′ → θ′ → Al_2_Cu. As shown in Figure 10a,d, there are many G–P zones and precipitated θ″ phases in the T6-treated alloy without TMF. According to the Gibbs–Thomson effect [25,26], the particle radius affects the solute concentration in the matrix adjacent to the particle. The solute concentration near small particles is higher than around large particles. The solute moves from small particles to large particles because of the concentration difference between different grains. After elevated-temperature TMF, the nano-precipitation phases (θ″ phases and G–P zones) transform into spherical particles (θ-Al_2_Cu phases), which is known as Ostwald coarsening or ripening [26,27]. Under the action of higher-temperature TMF (120–425 °C), the approximately rectangular massive precipitation phases (G–P zones and θ″ phases) change into spherical particles (θ phases). At the same time, the Al_2_Cu particles lose coherence with the matrix, leading to reduced strengthening effect and high-temperature cyclic softening (Figure 11b,d). The transformation of nano-precipitated phases (such as Al_2_Cu, θ′, and θ″) is related to time and the maximum cyclic temperature. With increasing the testing time, the precipitated phases of the UT alloy are continually coarsening, and the amount of precipitated θ phases clearly decreases for the UT alloy.

The damage mechanism of materials is influenced by cyclic deformation behavior, which determines the fatigue life. The fatigue damage normally shows a highly localized feature, which is probably derived from the magnification of fatigue damage distribution heterogeneity through the progressively cumulative process during cyclic loading [14,28]. Based on the earlier studies [29,30], the existing defects and micro-structures in materials affect the plastic deformation behaviors and fatigue damage distribution. The inhomogeneity of micro-structures causes corresponding stress or strain concentration, which may be directly related to the locality of fatigue damage [31]. From the perspective of micro-structures, the secondary phase particles play a significant role in the damage and deformation mechanisms of the Al–Si alloys under TMF. The fatigue fracture process can be divided into three sections: (1) nucleation of micro-cracks, (2) growth of micro-cracks, and (3) connection of micro-cracks. Any factor that provides assistance or resistance in each stage may also make a difference in fatigue life. For the multi-phase Al–Si alloy, the strength of the material mainly depends on dispersion strengthening (precipitated phases and primary phases). In the process of TMF, the consecutive thermal/mechanical cycling brings about the expansion mismatch between the hard phases and the ductile matrix, resulting in local microplastic deformation and the initiation of micro-cracks [32]. The slip introduced in the forward loading process does not recover fully in the reverse loading process, because of the interaction of particles and dislocations in materials. With the accumulation of non-reciprocal sliding, non-reciprocal plastic deformation shows the final strain localization, leading to crack initiation, in particular in the presence of brittle particles [31].

The nucleation of micro-cracks in particles basically differs in two aspects: (1) the calculation of the strain or stress concentration on particles due to pilling-up mechanism of dislocations, (2) the calculation of the critical strain or stress required for the cracking of particles or the interface decohesion between matrix and particles. The competing effects of the critical strain or stress for fatigue crack nucleation (crack initiation at fractured Si particles, interface between particles and matrix) can be found (Figure 10a,b). For the AC alloy, the dislocation accumulation mechanism is the most likely mechanism to explain the cracking of Si particles (Figure 10a). The stress acting on primary particles is higher for the large primary Si particles, resulting in the fracture of particles and initiation of micro-cracks by the action of strong stress concentration, due to the pilling-up of dislocations. The continued mechanical and thermal loads result in the networks of micro-cracks that eventually exceed the critical size and lead to fatigue fracture (Figure 10c). The micro-plastic deformation and creep around the interface are smaller, because of the significant damage by broken primary Si. A larger primary phase leads to irreversible plastic stress or strain concentration and limited critical stress or strain of crack under cyclic load, resulting in lower fatigue crack initiation life. As for decreasing the particle size below a critical size for the UT alloy, the critical strain or stress required for the cracking of particles increases. It is not easy to form fatigue cracks with breaking primary Si. However, the interface decohesion between particles and the matrix becomes another important crack initiation location. Given the fact that the highest temperature is about 0.823 Tm (the absolute melting point), it is much easier for micro-voids to form around the interface combined with micro-plasticity deformation and creep. These micro-pores are usually located near the interface between the particles and matrix, and finally coalesce to micro-cracks (Figure 10b). The failure mechanism is the interaction of thermal, mechanical, and thermo-mechanical loading, which leads to fatigue crack initiation and propagation. In sum, the depressed crack initiation and growth rate can significantly improve the fatigue life of the UT alloy. Generally speaking, larger Si grains in the UT alloy cause higher plastic strain and, eventually, fatigue failure. Nevertheless, the UT not only reduces the maximum size of primary Si, but also optimizes the configuration and 3D connection of the phases. Therefore, compared with the AC alloy, it can achieve a longer fatigue life. In sum, there are two different crack initiation mechanisms under TMF: one is primary Si fracture for the AC alloy, and the other is primary Si debonding for the UT alloy. After the crack initiation, the fractured or debonded primary phases provide the advantages for further development of main cracks for both alloys. As mentioned above, fatigue damage localization can lead to TMF damage, including micro-structure deformation, and fatigue crack nucleation and propagation. The extension of fatigue life is observably affected by the location of crack initiation and the fatigue crack propagation for the cast Al–Si alloys. The finer structures of the UT alloy have a positive impact on the TMF resistance, such as increasing the critical stress or strain required for crack initiation and propagation.

## 4. Conclusions

The TMF behaviors of the Al–Si piston alloys including cyclic behaviors, fatigue life, and failure mechanisms were studied. According to the experimental results, the following conclusions can be obtained:The grain size, primary Si, and intermetallic compound are reduced significantly in the UT alloy. The tensile, isothermal low-cycle fatigue and TMF resistance are significantly improved due to the amelioration of the micro-structures. The UT alloy (805 ± 253 cycles) has about twice the TMF life of the AC alloy (403 ± 98 cycles);As a result of pilling-up of dislocations, the competitive effects of the critical strain/stress for fatigue crack nucleation can be found: one is primary Si fracture for the AC alloy with limited critical strain/stress for fatigue crack nucleation at fractured Si particles, and the other is primary Si debonding for the UT alloy with increasing critical fracture strain/stress. The micro-cracks are perpendicular to the loading direction for both alloys;After the crack initiation, the fractured or debonded primary phases provide the favorable conditions for further development of main cracks along pre-existing damage. The close-packed primary Si and intermetallic particles are the major cause of limited critical stress or strain for fatigue crack initiation for the AC alloy. The finer structures of the UT alloy have a positive impact on the TMF resistance. Reduced volume fractions of highly interconnected rigid phases can lead to the deformation reversibility and homogeneity for the UT alloy. The damage behaviors change from concentrated damage near then main crack to rather dispersed damage with fractured or debonded Si.

## Figures and Tables

**Figure 1 materials-15-07113-f001:**
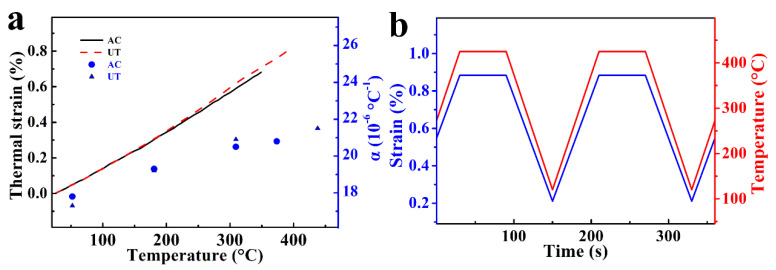
(**a**) The thermal expansion coefficient and the thermal stain; (**b**) IP loading condition of TMF.

**Figure 2 materials-15-07113-f002:**
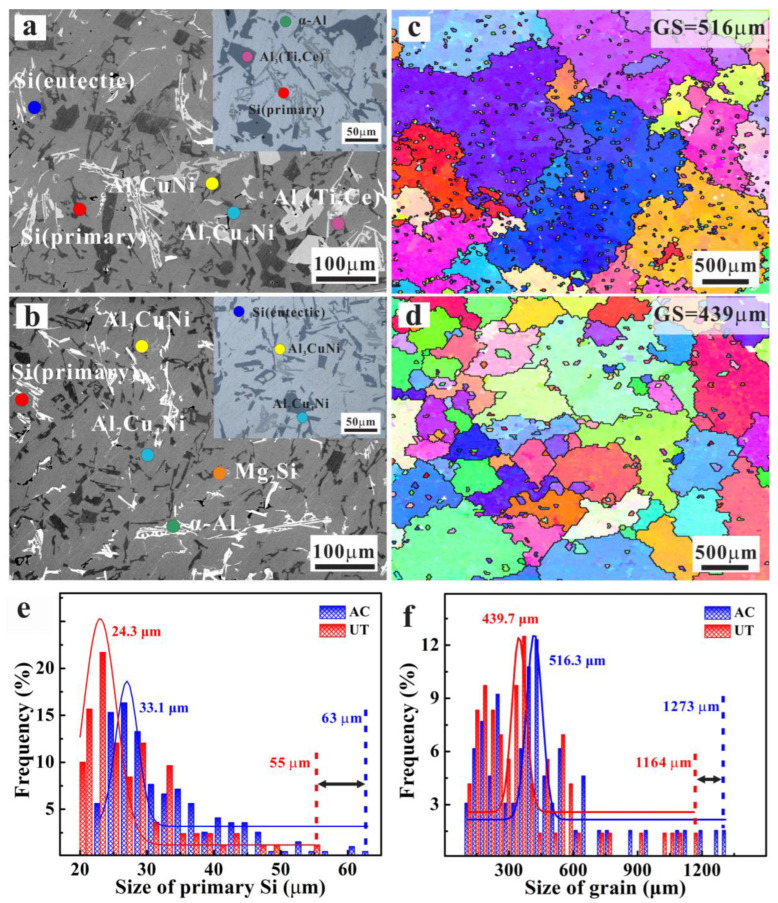
Typical microstructure analysis of the two Al–Si piston alloys: microstructures of AC alloy (**a**) and UT alloy (**b**); the EBSD images of AC alloy (**c**) and UT alloy (**d**); (**e**) size distribution of primary Si; (**f**) grain size distribution.

**Figure 3 materials-15-07113-f003:**
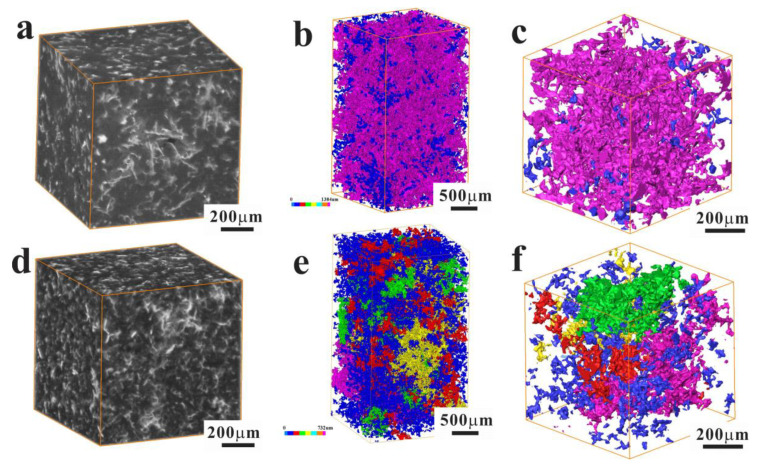
3D XRM images showing the microstructures of AC (**a**,**b**) and UT (**c**,**d**) alloys: (**a**,**d**) 3D visualization reconstruction; (**b**,**e**) the volume renderings of the intermetallic particles with different size; (**c**,**f**) the magnification of regions marked in (**b**,**e**).

**Figure 4 materials-15-07113-f004:**
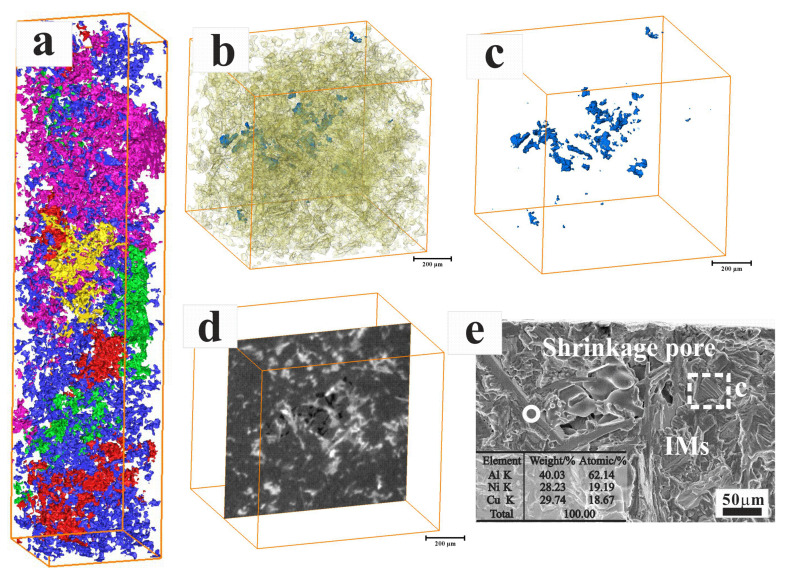
**The** 3D XRM images showing the micro-structures of 3D visualization reconstruction (**a**); the volume renderings of the intermetallic particles (in yellow, (**b**)) and shrinkage pores (in blue, (**c**)); (**d**) the intermetallic particles and shrinkage pores; (**e**) fatigue crack initiation in shrinkage pore.

**Figure 5 materials-15-07113-f005:**
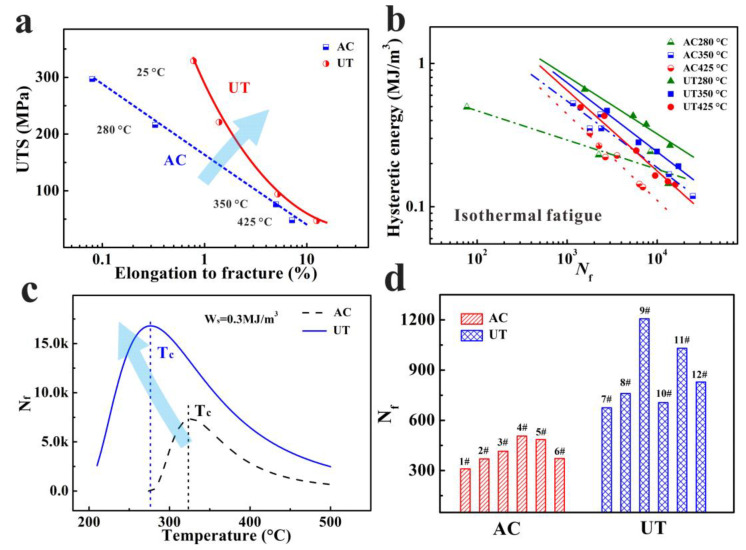
Properties of the two alloys (**a**) relation between the UTS and elongation-to-fracture; (**b**) low-cycle fatigue behaviors; (**c**) relationship of *N*_f_–*T* at a constant *W*s; (**d**) the TMF life of the two alloy at constraint factor (η = 1).

**Figure 6 materials-15-07113-f006:**
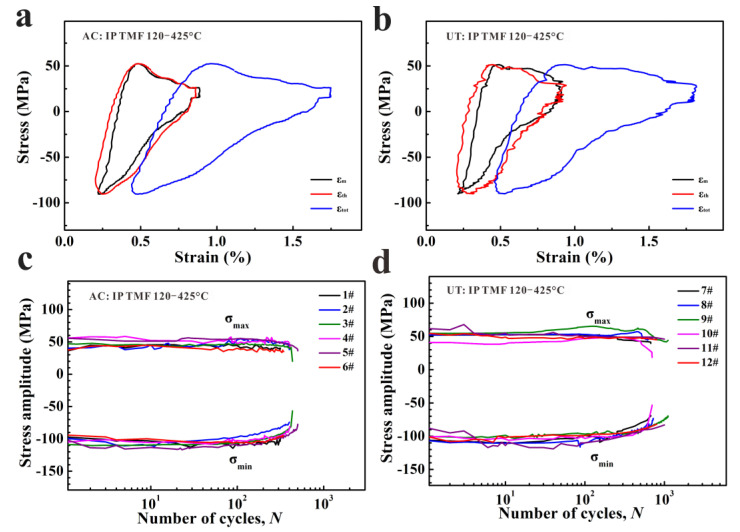
Cyclic deformation behaviors: (**a**,**b**) half-life hysteresis loops; (**c**,**d**) cyclic stress response curves.

**Figure 7 materials-15-07113-f007:**
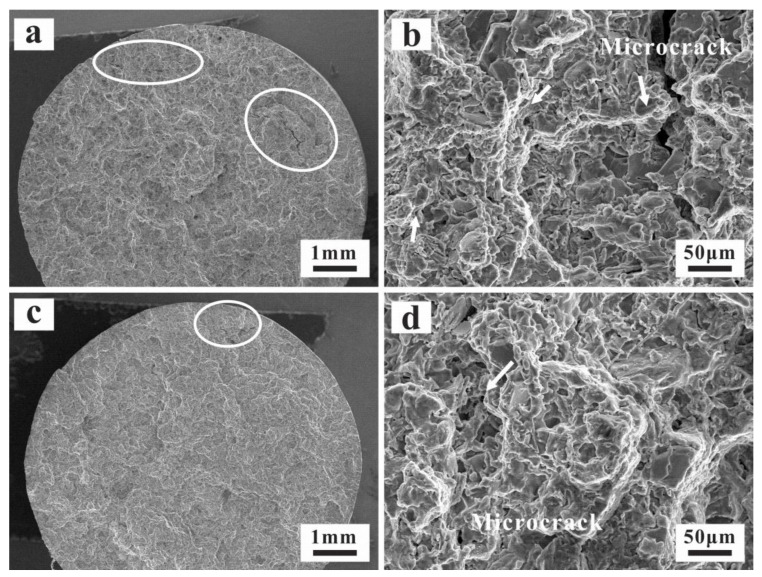
Fractography micrograph of fractured specimen: (**a**,**b**) AC; (**c**,**d**) UT.

**Figure 8 materials-15-07113-f008:**
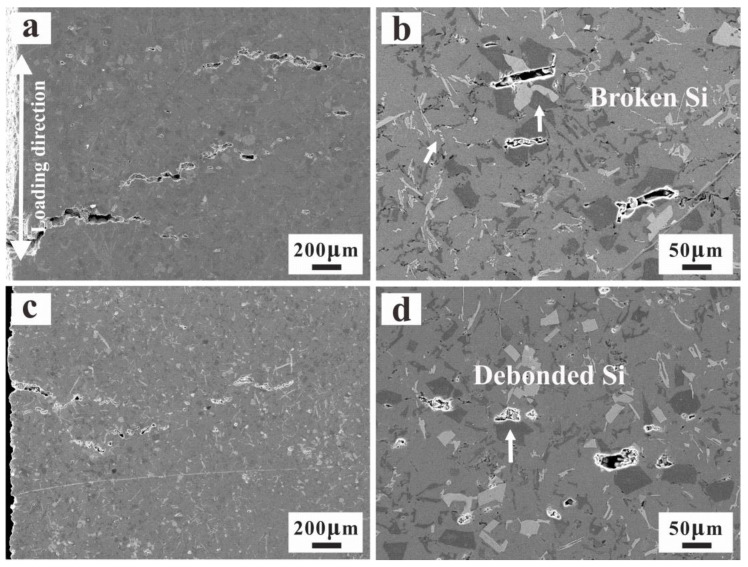
Microscopic damage characteristics around main fatigue crack of the two alloys at constraint factor (η = 1): (**a**,**b**) AC; (**c**,**d**) UT.

**Figure 9 materials-15-07113-f009:**
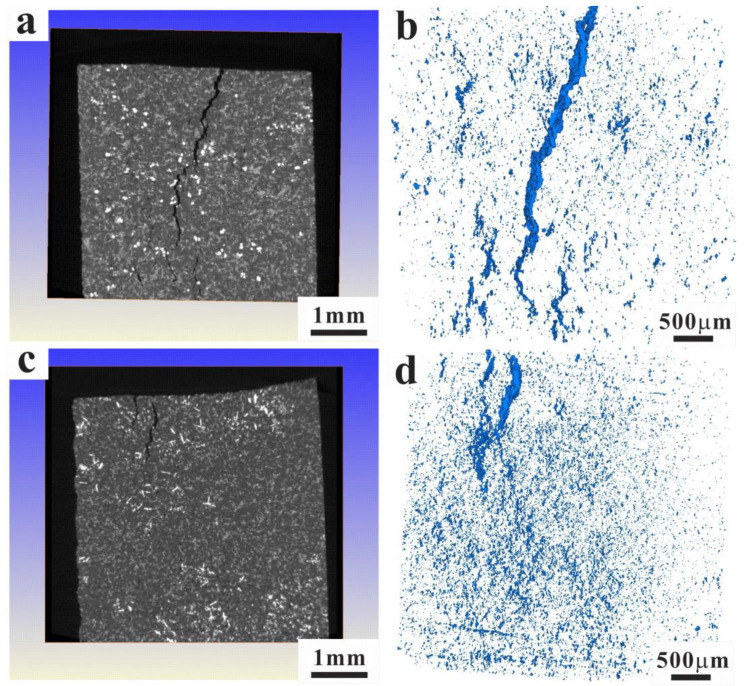
The 3D XRT images showing the main crack of AC (**a**,**b**) and UT (**c**,**d**) alloys after TMF.

**Figure 10 materials-15-07113-f010:**
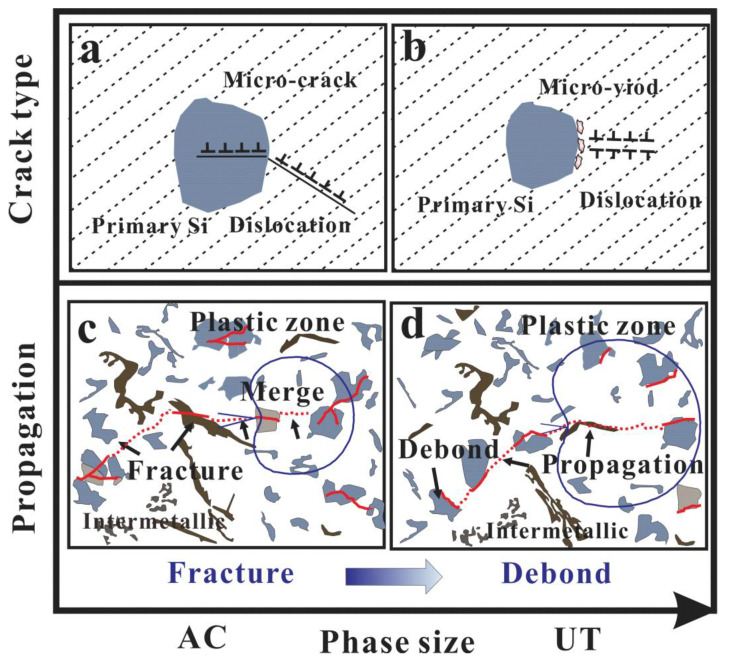
Schematic illustration of crack initiation and propagation mechanisms: fatigue crack initiation (**a**) and propagation mechanism (**c**) in fractured Si of AC alloy; fatigue crack initiation (**b**) and propagation mechanism (**d**) in debonded Si of UT alloy.

**Figure 11 materials-15-07113-f011:**
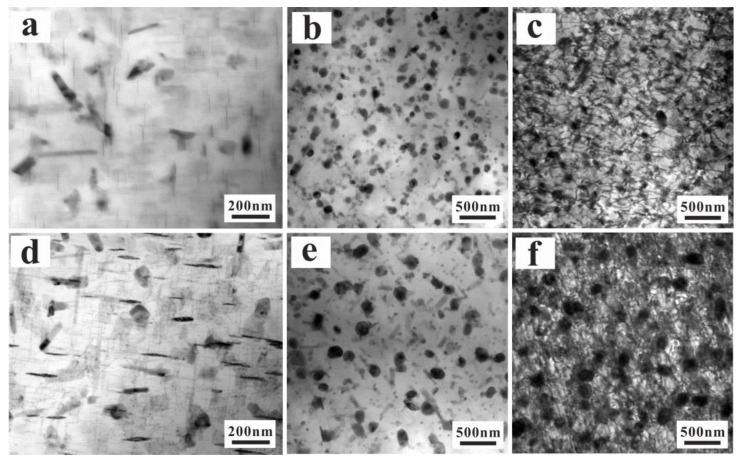
The TEM observation of the TMF samples: the microstructure of AC alloy before (**a**) and after TMF (**b**,**c**); the microstructure of UT alloy before (**d**) and after TMF (**e**,**f**).

**Table 1 materials-15-07113-t001:** Chemical composition (wt.%) of the Al–Si piston alloy.

Element	Si	Cu	Ni	Mg	Ti	Zn	Ce	Fe	V	Al
**AC (%)**	12.10	3.42	2.16	0.82	0.21	0.19	0.17	0.24	0.016	Bal.
**UT (%)**	11.81	3.52	2.26	0.85	0.20	0.18	0.18	0.26	0.017	Bal.

## Data Availability

Not applicable.

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
