# Peer review of "Optimization of Thermo-Mechanical Fatigue Life for Eutectic Al–Si Alloy by the Ultrasonic Melt Treatment"

_materials, 2022, doi:10.3390/ma15207113_

Round 1
Reviewer 1 Report
The paper deals with the interesting subject of thermo-mechanical fatigue life study of eutectic Al-Si alloys. The paper is interesting, but it should be improved to prove the correctness of achieved results:
1. Abstract - please add quantitative results.
2. The research methodology is not described:
- the method of strain measurement is not given,
- the fatigue test is not described - what is the shape and values of loading curve, how loading was applied,
- for XRM and SEM - please give the resolutions applied, the used apparatus,
- how the SEM and XRM points of observation were selected, how many observations were made,
- for all testing equipment state the accuracy of measurement,
- for fatigue tests - how many samples were tested, what dimensions and shape of samples were used, what is the results distribution, are they repeatable?
3. Results:
- fig. 6 needs an explanation - why in a and b the curve dose not start from 0 for strain, also the beginning of the chart should be in point (0,0) - please add the axis for stress = 0, what is the reason of the change in curves shapes in c and d. In fig d - please change "ampliture" to "amplitude",
- fig. 11 needs an improvement - please add the legend of the used colors and curves.
4. What is novel in the presented study - the literature review shows that there are some of such studies.
5. What are the future plans and the application possibilities of presented work.
Reviewer 2 Report
The paper presents a good (and interesting) technical report on improving the fatigue life of Al-Si eutectic alloys.
Some details have to be addressed before the paper can be recommended for publication.
English needs tidying – please, be careful in using the past, present, and future tense. Some terms have to be better chosen (e.g. ultrasonic agitation instead ultrasonic irradiation, uniformly distributed (?) instead uniformity, etc.).
More specific remarks:
What is the meaning of colors showing the intermetallics in figs. 3b,d, and 4a? Is the same as in fig. 2? This should be clearly stated in the legend of figures. How these phases were identified during the 3D XRM?
Paragraph 3.2 – not only UTS and EF but also YS (yield strength) is important as it incorporates the effect of grain size and particle spacing (especially GP zones and θ-phases). Please, provide this information, and discuss the effect of temperature and ultrasonic treatment on YS separately from UTS and EF.
Round 2
Reviewer 1 Report
The authors corrected the article taking into account all my comments. In my opinion the study can be published in the present form.